# Sarcopenic Factors May Have No Impact on Outcomes in Ovarian Cancer Patients

**DOI:** 10.3390/diagnostics9040206

**Published:** 2019-11-28

**Authors:** Naomi Nakayama, Kentaro Nakayama, Kohei Nakamura, Sultana Razia, Satoru Kyo

**Affiliations:** 1Faculty of Health and Nutrition, The University of Shimane, 155 Nishihayashigi, Izumo, Shimane 693-8550, Japan; 2Department of Obstetrics and Gynecology, Shimane University, 89-1 Enya, Izumo, Shimane 693-8501, Japan; kohei320@med.shimane-u.ac.jp (K.N.); raeedahmed@yahoo.com (S.R.); satoruky@med.shimane-u.ac.jp (S.K.)

**Keywords:** ovarian cancer, sarcopenia, skeletal muscle index, intramuscular adipose tissue content

## Abstract

Although the prognostic value of sarcopenic factors, such as loss of muscle mass and quality, have been widely reported in patients with cancer during the last decade, the value in those with ovarian cancer remains unclear. Therefore, this study evaluated the prognostic impact of sarcopenic factors in patients with ovarian cancer. We retrospectively evaluated the data of 94 ovarian cancer patients who underwent surgery and chemotherapy at the Shimane University Hospital between March 2006 and 2013. Preoperative computed tomography scan at the level of the third lumbar vertebra was used to evaluate skeletal muscle volume and quality based on the skeletal muscle index (SMI) and intramuscular adipose tissue content (IMAC), respectively. The impact of preoperative SMI and IMAC on outcomes was subsequently investigated. Low SMI and high IMAC were not significantly associated with disease-free survival (*p* = 0.329 and *p* = 0.3370, respectively) or poor overall survival (*p* = 0.921 and *p* = 0.988, respectively). Neither preoperative low muscle volume nor low muscle quality was a poor prognostic factor in ovarian cancer.

## 1. Introduction

Ovarian cancer is the most lethal gynecological malignancy worldwide [1]; its incidence has markedly increased in the last decade. Owing to the lack of specific symptoms and effective screening modalities, the majority of patients have peritoneal dissemination and distant metastases at the time of diagnosis. Although survival has markedly improved after the introduction of platinum–taxane combination chemotherapy, the overall five-year survival remains around 45%. The International Federation of Gynecology and Obstetrics stage and residual tumor volumes are well-known prognostic factors for ovarian cancer; however, these are unmodifiable at the time of diagnosis [2]. Prognostic factors that may be modified through supportive care are the key to improving prognosis.

Sarcopenia was initially described as an age-related phenomenon of loss of skeletal muscle mass [3]. However, sarcopenia is currently defined as a syndrome characterized by progressive loss of skeletal muscle mass and quality, and many studies have reported significant associations between sarcopenia and poor outcomes in various kinds of diseases, including cancer [4,5,6,7,8,9]. In particular, sarcopenic factors are reportedly associated with the prognosis of digestive organ cancers, such as hepatocellular carcinoma (HCC) [10], pancreatic cancer [11], and biliary duct [12] and gastric cancer [13]. Sarcopenia is known to be modifiable by proper nutritional interventions and physical exercise. Nutritional and rehabilitative interventions have, therefore, been recommended both before and during cancer treatment to improve prognosis. There are several reports on the impact of sarcopenic factors on the outcomes in patients with ovarian cancer. However, the findings are inconsistent, and there is no consensus regarding the relationship between sarcopenic factors (i.e., skeletal muscle quantity and quality) and patient prognosis in ovarian cancer [14,15]. Therefore, the present study aimed to evaluate the sarcopenic factors by using cut-off values established in the same ethnic population; it also aimed to analyze its impact on the outcomes of patients with ovarian cancer in the Japanese population.

## 2. Materials and Methods 

### 2.1. Patients

We retrospectively evaluated 94 patients with ovarian cancer, who were treated at the Shimane University Hospital between March 2006 and 2013. All patients were primarily treated surgically (with total abdominal hysterectomy, bilateral salpingo-oophorectomy, and omentectomy with or without pelvic and para-aortic lymph node dissection) and adjuvant taxane-platinum combination chemotherapy. They underwent preoperative plain CT at the level of the third lumbar vertebra (L3). CT was taken within one week before surgery. This study was approved by the Ethics Committee of Shimane University (IRB No. 20070305-1, 20070305-2, 22 September 2018) and was conducted in accordance with the 1996 Declaration of Helsinki.

### 2.2. Image Analysis

Cross-sectional unenhanced CT images of the L3 level were used to evaluate skeletal muscle and adipose tissue. The skeletal muscle area consisted of the psoas, paraspinal (erector spinae, multifidus, and quadratus lumborum), and abdominal wall muscles (transversus abdominus, external and internal obliques, and rectus abdominus). Skeletal muscle, visceral adipose tissue, and subcutaneous adipose tissue were identified and quantified according to Hounsfield unit (HU) thresholds of −29 to 150, −150 to −50, and −190 to −30 HU, respectively. Skeletal muscle quantity was evaluated based on the SMI, calculated by normalizing the cross-sectional images of muscle area to the height of the patient in meters squared. Skeletal muscle quality was evaluated according to IMAC, calculated by dividing the CT attenuation of the erector spinae and multifidus muscles (HU) by that of the subcutaneous adipose tissue (HU). Low SMI was regarded as a proxy for low muscle mass, and high IMAC was considered to indicate low muscle quality. IMAC was used in several scientific reports and high IMAC was identified as an independent risk factor for poor outcomes after living donor liver transplantation (LDLT) [4], resection of hepatocellular carcinoma [10], pancreatic cancer [11], and extrahepatic biliary malignancies [16].

Till date, sarcopenia working groups in Europe and Asia have not proposed cut-off values for sarcopenia determined via CT; however, certain diagnostic cut-off values have been proposed using bioelectrical impedance analyses and dual X-ray absorptiometry. We, therefore, used the cutoff values recently established by Kaido et al. based on the data of 657 Japanese healthy individuals [17]. The sex-specific cut-off value for low SMI was defined as more than two standard deviations (SDs) below the mean SMI of healthy individuals (<50 years), while high IMAC was defined as more than the two SDs above the mean IMAC of healthy individuals (<50 years). Since all ovarian cancer patients are female, we only used the cutoff values of SMI and IMAC for females (30.88 and −0.229, respectively). Normal/low SMI and normal/high IMAC were defined based on these values. We then defined low SMI as reduced muscle mass and high IMAC as reduced muscle quality. The IMAC of most ovarian cancer patients in this study was lower than the cutoff values used for other types of cancer. Therefore, we used the median IMAC (−0.511) as a cutoff value in this study.

### 2.3. Analyzed Parameters

The clinicopathological characteristics of the patients classified according to SMI and IMAC were analyzed on the basis of the following variables: Age, BMI, tumor markers including carcinoembryonic antigen, carbohydrate antigen 19-9, and sialyl-Tn, FIGO stage, postoperative complications, preoperative SMI and visceral fat, and length of hospital stay. The OS and DFS were analyzed based on the preoperative SMI and IMAC. 

### 2.4. Statistical Analyses

Statistical analyses were performed using the SPSS 24.0 (IBM Corporation, Armonk, NY, USA) software. A *p*-value of <0.05 was considered statistically significant. Differences between groups were evaluated using Student’s *t* and χ^2^ tests for continuous and categorical variables, respectively. The DFS and OS in the subgroups were compared using Kaplan-Meier curves and log-rank tests. Univariate analyses were conducted to identify factors significantly associated with patient survival, and their hazard ratios and 95% confidence intervals were calculated.

## 3. Results

Among the 94 patients included in this study, 48 had advanced-stage disease (FIGO stage III and IV). The patient characteristics are shown in Table 1. The mean body mass index (BMI) at diagnosis was 22.9 ± 3.7 kg/m^2^. Data on the sarcopenic factors at diagnosis are shown in Table 2. The median skeletal muscle index (SMI) and intramuscular adipose tissue content (IMAC) were 34.93 (range, 18.33 to 54.64) and −0.511 (range, −1.120 to −0.23), respectively. The clinic-demographic factors were classified according to the presence of reduced muscle mass and quality in the preoperative period (Table 2 and Table 3). Although sarcopenia was more prevalent among older people, age was not associated with muscle mass and quality. There was also no significant association between sarcopenic factors and tumor markers, FIGO stage, postoperative complications, and length of hospital stay. 

The overall survival (OS) and disease-free survival (DFS) rates based on skeletal muscle mass and quality are summarized in Figure 1a,b and Figure 2a,b). We found no significant difference in terms of OS and DFS when patients were classified based on skeletal muscle mass (*p* = 0.3370 (Figure 1a)) and *p* = 0.329 (Figure 1b), respectively) and muscle quality (*p* = 0.988 (Figure 2a) and *p* = 0.921 (Figure 2b), respectively). 

## 4. Discussion 

Sarcopenic factors have been reported to influence cancer prognoses. Although there have been several reports regarding patients with ovarian cancer, the results of these studies are inconsistent. This may be partly attributed to the lack of uniformity in the cut-off used to define sarcopenia on computed tomography (CT) images; this would directly influence the statistical results. Selection bias caused by the difference in patients’ clinical stage is also likely to affect the results [18,19]. Therefore, we evaluated the sarcopenic status in the Japanese population using cut-off values established and used specifically in this ethnic population. This retrospective study showed that preoperative quantity and quality of skeletal muscle mass were not associated with poor prognoses among ovarian cancer patients; however, these parameters have been reported to be prognostic in patients with HCC [10], pancreatic cancer [11], biliary duct cancer [12], and gastric cancer [12]. Therefore, we were particularly interested in evaluating the possible differences in outcomes between these reports and the present study. We hypothesized that any difference may be attributable to the varying propensity for sarcopenia between the different cancer types. Sarcopenia may be categorized into two types, namely, primary sarcopenia induced by aging and secondary sarcopenia induced by several diseases that accompany chronic inflammation [20]. There are several causes of sarcopenia in cancer patients, and its degree differs based on the patient’s susceptibility. In general, ovarian cancer patients are less susceptible to sarcopenia than those with other cancers. In this study, the IMAC of ovarian cancer patients is considerably lower than that of those with other types of cancer, such as HCC and pancreatic cancer [10,11]. We speculated that muscle quality is preserved in ovarian cancer patients as they have a lower susceptibility to sarcopenia owing to certain reasons. First, ovarian cancer patients are all women, and the age-related decrease in muscle quantity and quality is much smaller in women than in men [21] as the age-related decrease in muscle quantity and quality is considerably lower in women than in men. Studies that investigated the impact of sarcopenia on cancer prognosis did not consider patients’ sex; therefore, there are no sex-specific data. However, there could be a sex difference in the impact of sarcopenia on cancer prognosis. Second, gastrointestinal (GI) symptoms reduce oral food intake in cancer patients, causing malnutrition-induced sarcopenia. GI symptoms are more frequent in patients with GI and hepatobiliary and pancreatic cancer than in those with other cancers, including ovarian cancer [22]. While ovarian cancer symptoms are often unspecific, they rarely include GI symptoms [23]. Ovarian cancer itself usually does not affect patient appetite and oral intake until the advanced stages. Third, cancer cachexia is a complex condition of tissue wasting that develops as a secondary disorder in cancer patients and leads to progressive functional impairment [24]. It significantly affects the skeletal muscle and causes its wasting in cancer patients. Therefore, cachexia is considered to be a major cause of secondary sarcopenia in cancer patients. The prevalence of cachexia varies according to the type of cancer, with the prevalence being higher in GI, liver, and pancreatic cancers than in other sites (40–80% versus only 0.5%) [24]. Additionally, cachexia is less prevalent in ovarian cancer than in cancers of the digestive organs. It is more prevalent in cancers whose prognosis is influenced by sarcopenia. We speculated that sarcopenia at the time of cancer diagnosis reflects the progression of cachexia, indicating a poorer prognosis.

Sarcopenia may be modified through proper nutritional care and rehabilitation. The importance of evaluating sarcopenia and providing appropriate interventions in cancer-related cases has been highlighted in certain types of cancer [8,9,10,11,12,13]. Although we found no association between sarcopenic factors and the prognosis of ovarian cancer, the body composition of ovarian cancer patients should be carefully assessed as sarcopenia may occur secondary to invasive cancer treatment, such as surgery and chemotherapy. Certain studies have shown that skeletal muscle mass decreases during invasive treatment [16,25]. In the current study, all patients had undergone curative or volume reduction surgery followed by chemotherapy. We speculated that these treatments would affect the SMI and IMAC of ovarian cancer patients. In addition, the prevalence of cachexia generally increases with advancing clinical stage [24]. In this regard, the impact of sarcopenic factors on prognosis may vary across different time points throughout the disease course. 

Therefore, nutritional support and rehabilitation are among the important supportive interventions for cancer. Gagnon et al. reported that interdisciplinary nutrition–rehabilitation programs may improve the well-being of cancer patients and should be considered an integral part of standard care for these patients [26]. Exercise, aimed at maintaining bone and muscle, is recommended for a better quality of life (QOL) in patients undergoing treatment for cancer [27,28]. Resistance and aerobic exercises have been shown to preserve or improve bone density and contribute to better QOL in cancer survivors and in patients actively undergoing hormone therapy [17,29,30,31,32]. Dietary counseling and nutritional support have also been reported to positively influence morbidity outcomes and QOL in cancer patients undergoing radiation therapy [33]. Early consultation with a skilled nutritionist is widely accepted to be beneficial for cancer patients receiving anticancer treatments [18,19,34,35,36] and for those in advanced stages of malignancy [37]. Collectively, these findings indicate that clinicians should carefully evaluate patients’ nutritional conditions, with particular emphasis on the presence of sarcopenia, and consider nutritional interventions throughout the disease course. 

This study has certain limitations. First, the definition of sarcopenia diagnosed via CT, has varied in previous studies [6,7,8,9,10,11,12,13]. There is different evaluation method of myosteatosis besides IMAC, such as skeletal muscle radiation attenuation (SMRA) [38]. The criteria and evaluation method used in this study need to be recognized as subject to further validation and even expansion. Second, this study included patients from all clinical stages ranging from I to IV; this heterogeneity may have affected the results. Finally, this was a retrospective single-institution study, with a relatively small sample size. The findings should, therefore, be validated in larger prospective cohorts. 

In conclusion, reduced preoperative muscle mass and quality may not be prognostic factors in patients with ovarian cancer. However, sarcopenia may occur during the treatment of any type of cancer; the patients’ body composition should, therefore, be closely monitored, and nutritional and rehabilitative interventions should be provided when needed.

## Figures and Tables

**Figure 1 diagnostics-09-00206-f001:**
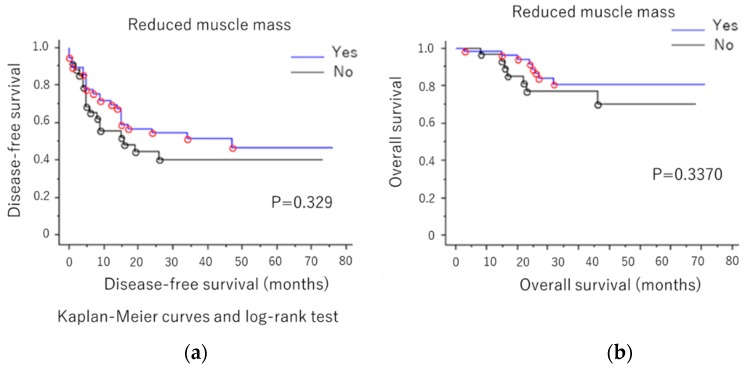
(**a**) Disease-free survival (DFS) and (**b**) overall survival (OS) rates according to muscle mass.

**Figure 2 diagnostics-09-00206-f002:**
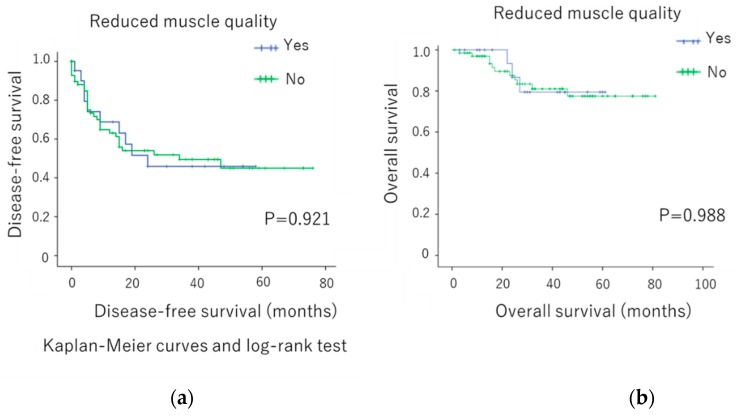
(**a**) Disease-free survival and (**b**) overall survival rates according to muscle quality.

**Table 1 diagnostics-09-00206-t001:** Characteristics of patients.

Characteristics	Mean ± SD (%)	Characteristics	Number of Patients (%)
**Age (Years)**	61.8 (range: 25–84)	**FIGO stage**	
**Weight (kg)**	53 ± 8.6	I	37 (39.4)
**Height (m)**	1.5 ± 0.06	II	9 (9.6)
**BMI (kg/m^2^)**	22.9 ± 3.7	III	30 (31,9)
**Initial tumor marker**		IV	18 (19.1)
CA125	1250.5 ± 2659.7	**Histology**	
CEA	15.9 ± 89.9	Serous	45 (47.8)
STN	216.1 ± 569.6	Endometrioid	21 (22.3)
CA19-9	448.4 ± 2898.1	Mucinous	12 (12.8)
		Clear cell	16 (17.1)
		Other	0 (0)
		**Tumor grade**	
		Grade 1	11 (11.7)
		Grade 2	32 (34)
		Grade 3 (Clear cell included)	51 (54.2)
		**Residual tumor**	
		Positive	45 (47.9)
		Negative	49 (52.1)
		**Recurrence**	
		Yes	44 (46.8)
		No	50 (53.2)

**Table 2 diagnostics-09-00206-t002:** Clinicopathological characteristics of the patients classified according to skeletal muscle index (SMI).

	Reduced Muscle Mass		
	No (*n* = 32)	Yes (*n* = 62)	*p*-Value
Age	61.2 ± 10.5	61.6 ± 12.9	*p* = 0.838
BMI	24.9 ± 5.8	21.5 ± 2.8	*P* = 0.005
CA125	1005.0 ± 1822.2	1379.4 ± 3014.7	*p* = 0.493
STN	259.6 ± 864.0	193.4 ± 338.4	*p* = 0.018
CA19-9	203.4 ± 364.7	600.9 ± 3687.9	*p* = 0.488
CEA	32.2 ± 141.2	4.96 ± 8.65	*p* = 0.315
FIGO stage III, IV (%)	14/32(43.7%)	34/62(54.8%)	*p* = 0.3851
SMI	42.6 ± 4.4	31.6 ± 4.2	*p* = 0.000
Visceral fat	106.5 ± 65.2	52.0 ± 38.5	*p* = 0.317
Postoperative complication (%)	5/32(15.6%)	14/62(22.5%)	*p* = 0.5892
Length of stay (days)	17.4 ± 8.7	18.0 ± 10.3	*p* = 0.766

Student’s *t* and χ^2^ test.

**Table 3 diagnostics-09-00206-t003:** Clinicopathological characteristics of the patients classified according to intramuscular adipose tissue content (IMAC).

	Reduced Muscle Quality		
	No (*n* = 73)	Yes (*n* = 21)	*p*-Value
Age	62.0 ± 12.0	62.6 ± 11.4	*p* = 0.582
BMI	22.3 ± 3.4	23.5 ± 6.5	*p* = 0.418
CA125	1410.2 ± 2973.1	636.8 ± 1283.5	*p* = 0.101
STN	207.5 ± 618.5	175.2 ± 298.3	*p* = 0.808
CA19-9	130.1 ± 281.8	1435.4 ± 5827.5	*p* = 0.352
CEA	18.8 ± 104.0	5.7 ± 10.9	*p* = 0.389
FIGO stage III, IV (%)	38/73 (52.0%)	10/21 (47.6%)	*p* = 0.8065
SMI	34.3 ± 5.7	39.0 ± 8.4	*p* = 0.028
Visceral fat	54.65 ± 40.7	126.7 ± 64.1	*p* = 0.000
Postoperative complication (%)	17/73(23.2%)	2/21(9.5%)	*p* = 0.2251
Length of stay (days)	17.4 ± 8.7	18.0 ± 10.3	*p* = 0.47

Student’s *t* and χ^2^ test.

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
