# Peer review of "Sarcopenic Factors May Have No Impact on Outcomes in Ovarian Cancer Patients"

_diagnostics, 2019, doi:10.3390/diagnostics9040206_

Round 1
Reviewer 1 Report
This is an interesting study investigating the prognostic value of sarcopenia in ovarian cancer. This is a well-written manuscript. Although previous studies have addressed such topic, the information from Asia is lacking in this field. It should be mentioned that the body composition may vary among geographical regions, and ethnicity; it is relevant to have such information in clinical practice. However, there are several background and methodological weaknesses that if fixed, could improve the paper.
1. The prognostic value of sarcopenia or myosteatosis in ovarian cancer has been evaluated in previous studies(Aust et al. PLoS One. 2015 Oct 12;10(10):e0140403; Kumar et al. Gynecol Oncol. 2016 Aug;142(2):311-6; Rutten et al. J Cachexia Sarcopenia Muscle. 2016 Sep;7(4):458-66; Bronger et al. Int J Gynecol Cancer. 2017 Feb;27(2):223-232; Rutten et al. Eur J Surg Oncol. 2017 Apr;43(4):717-724; Ataseven et al. Ann Surg Oncol. 2018 Oct;25(11):3372-3379; Ubachs et al. J Cachexia Sarcopenia Muscle. 2019 Aug 7. doi: 10.1002/jcsm.12468.). I suggest the authors to introduce this background and discuss your findings with these studies. Thorough revise in the Introduction and Discussion should be made. The first sentence in your Discussion stated "Although sarcopenic factors have been reported to influence cancer prognosis, this was not investigated in ovarian cancer." should also be revised.
2. It is unclear why the authors used the IMAC as the index of skeletal muscle quality rather than skeletal muscle radiotherapy (SMD)? Previous studies evaluated the effects of skeletal muscle by skeletal muscle index and SMD, I strongly suggest the author to analyze the SMD to make this study comparable to previous studies mentioned above. This information would also useful in clinical practice and future studies in this field.
3. Ovarian cancer was surgically staged based on FIGO staging system. FIGO staging system is the most commonly used staging system for gynecological cancer, it's unclear why the authors used "Clinical stage"? How was your patients stages?
4. The authors should briefly introduce the treatment policies of ovarian cancer for these patients. Primary debulking surgery followed by adjuvant platinum-based chemotherapy or Neoadjuvant chemotherapy followed by interval dulking surgery? (Vergote et al. N Engl J Med 2010;363:943-53; Katsumata et al. Lancet Oncol. 2013 Sep;14(10):1020-6).
5. Your cohort included Clinical stage I-IV; it's a heterogeneous group of patients you analyzed and may affect your survival analysis and interpretation for the effect of sarcopenia in ovarian cancer. Furthermore, your number of patients was small so that the statistical power to detect differences in such heterogeneous group may be limited.
6. Since your study has heterogeneous "Clinical stage" and small number of patients, the conclusion may be overstated. The author should revise the conclusion as well as the Title.
In the Table 1:
Several errors should be corrected to improve the integraty of this Table.
a. initial tumor marker -> Initial tumor marker
b. Clinical TNM stage: there is no space between TNM and stage.
c. residual tumor -> Residual tumor; and the number (%) of negative should be 49 (52.1), your current Table showed 49 52.1).
d. You only stated the values in this Table were Mean ± SD (%)[Range). What's the meaning of numbers in terms of Clinical TNM stage, Histology, Tumor grade, Residual tumor, and Recurrence? You should describe the meaning in this table.
In the Table 2:
Since you had stated the median and range of these indexes in your manuscript. Why does you make this Table to show the same thing?
In the Table 3 and Table 4: Please check the typo error in the Table 3 and Table 4 like Table 1.
Author Response
We sincerely appreciate the careful review of our manuscript and the helpful suggestions, which have provided further insight into our study and have contributed considerably to its improvement.
Reviewer 1
Comment 1:
The prognostic value of sarcopenia or myosteatosis in ovarian cancer has been evaluated in previous studie. I suggest the authors to introduce this background and discuss your findings with these studies. Thorough revise in the Introduction and Discussion should be made. The first sentence in your Discussion stated "Although sarcopenic factors have been reported to influence cancer prognosis, this was not investigated in ovarian cancer." should also be revised.
Response:
We appreciate the pertinent observations and helpful suggestions.
We have referred to certain suggested reports and have revised the Introduction and Discussion sections accordingly.
We have added the underlined sentences to the Introduction (on page 2: lines 60-66) and Discussion (on page 3: lines 86-92) sections.
Comment 2:
It is unclear why the authors used the IMAC as the index of skeletal muscle quality rather than skeletal muscle radiotherapy (SMD)? Previous studies evaluated the effects of skeletal muscle by skeletal muscle index and SMD, I strongly suggest the author to analyze the SMD to make this study comparable to previous studies mentioned above. This information would also useful in clinical practice and future studies in this field.
Response:
We appreciate the valuable suggestion. As correctly observed, the skeletal muscle radiation attenuation, which is defined by the mean Hounsfield unit value of the skeletal muscle, was used to measure muscle quality in several previous reports. The IMAC was used to assess muscle quality in this study owing to certain reasons. The IMAC represents the quantity of infiltrated intramuscular adipose tissue and is calculated by dividing the CT attenuation (HU) of the erector spine and multifidus muscles by that of the subcutaneous adipose tissue. It is known to reflect individual muscle quality and has been used in several previous studies. The value may be normalized by individually dividing the attenuation of adipose tissue (HU). In addition, a gender specific cut-off point, which was established using the data of healthy Japanese individuals, is available; however, a standardized cut-off point of the CT attenuation for decreased muscle quality and quantity has not been established. In view of these factors, the SMI and SMD were not use in this study, and the IMAC was used instead. However, the universal method of evaluating muscle quality by CT images is acceptable; the cut-off point should be defined after considering the race and gender.
Comment 3.
Ovarian cancer was surgically staged based on FIGO staging system. FIGO staging system is the most commonly used staging system for gynecological cancer, it's unclear why the authors used "Clinical stage"? How was your patients stages?
Response:
We appreciate your pertinent observations. The diagnoses were made based on conventional histopathologic examination of sections stained with hematoxylin and eosin. The tumors were categorized according to the World Health Organization subtype criteria and were staged according to the International Federation of Gynecology and Obstetrics classification system. This has been clarified in the text.
Comment 4.
The authors should briefly introduce the treatment policies of ovarian cancer for these patients. Primary debulking surgery followed by adjuvant platinum-based chemotherapy or Neoadjuvant chemotherapy followed by interval dulking surgery? (Vergote et al. N Engl J Med 2010;363:943-53; Katsumata et al. Lancet Oncol. 2013 Sep;14(10):1020-6).
Response:
We appreciate your suggestion. All patients were primarily treated surgically (with total abdominal hysterectomy, bilateral salpingo-oophorectomy, and omentectomy with or without pelvic and para-aortic lymph node dissection) and adjuvant taxane/platinum combination chemotherapy. We have added this information to our manuscript on page 5, line 167-170.
Comment 5.
Your cohort included Clinical stage I-IV; it's a heterogeneous group of patients you analyzed and may affect your survival analysis and interpretation for the effect of sarcopenia in ovarian cancer. Furthermore, your number of patients was small so that the statistical power to detect differences in such heterogeneous group may be limited.
Comment 6.
Since your study has heterogeneous "Clinical stage" and small number of patients, the conclusion may be overstated. The author should revise the conclusion as well as the Title.
Response to comment 5 and 6
We appreciate the valuable suggestions.
We have revised our manuscript and have added a few sentences to page 5, lines 155-161 regarding the limitations of this study.
We have also changed the title.
In the Table 1:
Several errors should be corrected to improve the integraty of this Table.
initial tumor marker -> Initial tumor marker Clinical TNM stage: there is no space between TNM and stage. residual tumor -> Residual tumor; and the number (%) of negative should be 49 (52.1), your current Table showed 49 52.1). You only stated the values in this Table were Mean ± SD (%)[Range). What's the meaning of numbers in terms of Clinical TNM stage, Histology, Tumor grade, Residual tumor, and Recurrence? You should describe the meaning in this table.Response
We sincerely appreciate your suggestions and have revised the table accordingly.
In the Table 2:
Since you had stated the median and range of these indexes in your manuscript. Why does you make this Table to show the same thing?
Response
We appreciate your observations and have removed Table 2 to avoid repetition of data already presented in the text.
In the Table 3 and Table 4:
Please check the typo error in the Table 3 and Table 4 like Table 1.
Response
We sincerely appreciate your pertinent observations and have corrected the typographical errors in the tables.
Reviewer 2 Report
This study evaluated the impact of preoperative sarcopenic factors on survival in patients with ovarian cancer.
The authors concluded that sarcopenic factors had no prognostic significance in their patient's cohort.
There are several issues to be resolved.
1. I wonder that The number of included patients were too small to extract meaningful conclusions. How did the authors define the sample size (although a retrospective study) in their study ?
2. The duration between the days of CT and the operation date would be important because the longer duration might cause additional loss of muscle or vice versa. What is the mean or median days from CT taken to the operation date ?
3. Please describe the distribution of SMI or skeletal muscle quality using histogram and 4 quadrant distribution table(min, 25 percentile, median, 75 percentile and maximum).
4. In Table 3 and 4, included number of patients were different as 94 patients in Table 3 versus 90 patients in Table 4.
Please explain this.
5. In Table 3 and Table 4, the authors should include distribution of all stages (not just stage III).
6. In the discussion section,
Page 5 : line 105 - line 129 -> The authors described that sarcopenia can be modified using nutritional care or exercise based rehabilitation.
However, if we agreed that nutritional support or exercise can overcome the sarcopenia status, how these comments related with the conclusion?
According to the conclusion in this study, sarcopenia was not related with the survival. Thus this meant that preoperative sarcopenic status had no clinical importance.
Then why do we need to try to change the sarcopenic status to normal ?
The authors need to modify the discussion section for logical consistency.
Author Response
We sincerely appreciate the careful review of our manuscript and the helpful suggestions, which have provided further insight into our study and have contributed considerably to its improvement.
Reviewer2
This study evaluated the impact of preoperative sarcopenic factors on survival in patients with ovarian cancer.
The authors concluded that sarcopenic factors had no prognostic significance in their patient's cohort.
There are several issues to be resolved.
Comment 1
I wonder that The number of included patients were too small to extract meaningful conclusions. How did the authors define the sample size (although a retrospective study) in their study ?
Response
We sincerely appreciate your constructive criticism. As correctly observed, the sample size was relatively small; this may be attributed to the retrospective design and the fact that it was conducted in a single institution. We included all patients who were available at our institution for this study. We have mentioned this in the discussion section, stating that our findings should be confirmed by further prospective studies with larger sample sizes.
Comment 2
The duration between the days of CT and the operation date would be important because the longer duration might cause additional loss of muscle or vice versa. What is the mean or median days from CT taken to the operation date ?
Response
We appreciate your pertinent question. The CT was taken within one week before surgery; this has been added to the manuscript on page 5, line 171-172.
Comment 3
Please describe the distribution of SMI or skeletal muscle quality using histogram and 4 quadrant distribution table(min, 25 percentile, median, 75 percentile and maximum).
Response
We appreciate your valuable suggestions. The ranges of SMI and IMAC have been described in Table 2. Histograms were not considered necessary for this study as they would not be representative of the results.
Comment 4
In Table 3 and 4, included number of patients were different as 94 patients in Table 3 versus 90 patients in Table 4.
Please explain this.
Response
We appreciate your observation and apologize for the error. The numbers have been corrected in Tables 2 and 3.
Comment 5
In Table 3 and Table 4, the authors should include distribution of all stages (not just stage III).
Response
We appreciate your suggestion. The provided values represent the percentage of patients with an advanced stage disease, equivalent to TMN stages â…¢ and â…£. We have corrected the typographical errors in Tables 2 and 3.
Comment 6
In the discussion section,
Page 5 : line 105 - line 129 -> The authors described that sarcopenia can be modified using nutritional care or exercise based rehabilitation.
However, if we agreed that nutritional support or exercise can overcome the sarcopenia status, how these comments related with the conclusion?
According to the conclusion in this study, sarcopenia was not related with the survival. Thus this meant that preoperative sarcopenic status had no clinical importance.
Then why do we need to try to change the sarcopenic status to normal ?
Response
We appreciate your pertinent observations. In this study, we evaluated the sarcopenic status at the time of diagnosis, prior to surgical treatment. While battling cancer, patients undergo invasive treatment including surgery, chemotherapy, and radiotherapy; all these cause sarcopenia. Patients also suffer from surgical stress and treatment-related malnutrition. We found that preoperative sarcopenia does not have an impact on patient prognosis. However, it is reasonable to evaluate whether the sarcopenic status at other time points, such as after surgery or during chemotherapy, has an impact on prognosis. In this regard, particular attention should be paid to patients’ nutrition and sarcopenic status throughout the process of cancer. We also intend to evaluate the impact of sarcopenia on prognosis at different time points.
Round 2
Reviewer 1 Report
The authors had addressed several comments from the reviewer. However, several concerns remained and many typo errors were also found in the Tables. Remaining concerns are as follows:
1. You may need to add references regarding IMAC in the methods. Because the SMD are worldwide used and most of previous studies evaluated myosteatosis by using SMD, your references could help the readership to understand your results more clearly. It may also help generalize the use of IMAT. In addition, the use of IMAC could also make your study incomparable to previous studies and constitute a limitation of your study. Since there has been many studies evaluating sarcopenia or myosteatosis in ovarian cancer (J Cachexia Sarcopenia Muscle. 2019. DOI: 10.1002/jcsm.12468.), it would be better to clarify all the points above.
2. Based on your manuscript in the method (page 6, line 551), concerns regarding staging system remained. Ovarian cancer is surgically staged based on FIGO staging system. It is the AJCC that use TNM to stage ovarian cancer rather than FIGO stage. It also remains unclear why the authors used the "Clinical stage" thoroughly in the manuscript. Since the ovarian cancer is surgically staged, the use of the term "clinical stage" is not appropriate (https://www.cancer.gov/publications/dictionaries/cancer-terms/def/clinical-stage). I would suggest the authors to favorably revise the manuscript to meet the basic requirements of SCI journal.
In the Table 1:
There remained some typo errors in this Table. It's the authors' responsibility to ensure the quality and correctness to meet the standard of SCI journal. In addition, ovarian cancer is surgically staged disease based on FIGO staging system, it would be nice to use "FIGO stage" in your study.
Several typo errors should be corrected as follows:
Clinical "TMN" stage -> Clinical "TNM" stage
Reccurrnce -> Recurrence
In the Table 2:
Please provide abbreviations below this Table. What is STN? I also found a typo error in this Table "omplication" -> "complication".
In the Table 3:
What is STN?
Author Response
We sincerely appreciate the careful review of our manuscript and the helpful suggestions, which have provided further insight into our study and have contributed considerably to its improvement.
Reviewer 1
Comment 1.
You may need to add references regarding IMAC in the methods. Because the SMD are worldwide used and most of previous studies evaluated myosteatosis by using SMD, your references could help the readership to understand your results more clearly. It may also help generalize the use of IMAT. In addition, the use of IMAC could also make your study incomparable to previous studies and constitute a limitation of your study. Since there has been many studies evaluating sarcopenia or myosteatosis in ovarian cancer (J Cachexia Sarcopenia Muscle. 2019. DOI: 10.1002/jcsm.12468.), it would be better to clarify all the points above.
Answer 1
Thank you for your comment. We added some references regarding IMAC in image analysis of materials and methods part in manuscript, page6 line 166-169. We also mentioned about a limitation of comparison in discussion part, in page 5 line 136-137.
Comment 2.
Based on your manuscript in the method (page 6, line 551), concerns regarding staging system remained. Ovarian cancer is surgically staged based on FIGO staging system. It is the AJCC that use TNM to stage ovarian cancer rather than FIGO stage. It also remains unclear why the authors used the "Clinical stage" thoroughly in the manuscript. Since the ovarian cancer is surgically staged, the use of the term "clinical stage" is not appropriate (https://www.cancer.gov/publications/dictionaries/cancer-terms/def/clinical-stage). I would suggest the authors to favorably revise the manuscript to meet the basic requirements of SCI journal.
Answer
Thank you very much for the comment.
We changed TNM to FIGO to meet the requirement of SCI journal as you suggested.
Comment 3
In the Table 1:
There remained some typo errors in this Table. It's the authors' responsibility to ensure the quality and correctness to meet the standard of SCI journal. In addition, ovarian cancer is surgically staged disease based on FIGO staging system, it would be nice to use "FIGO stage" in your study.
Several typo errors should be corrected as follows:
Clinical "TMN" stage -> Clinical "TNM" stage
Reccurrnce -> Recurrence
In the Table 2:
Please provide abbreviations below this Table. What is STN? I also found a typo error in this Table "omplication" -> "complication".
In the Table 3:
What is STN?
Answer 3
Thank you very much for your suggestion. We described what STN means in abbreviations part in manuscript. We also correct all typos.
Reviewer 2 Report
The manuscript has improved after revision
and it can be accepted in this form. All comments have been well addressed.
Author Response
Dear reviewer 2
We sincerely appreciate the careful review of our manuscript.
We think our manuscript is revised for the publication.
Thank you very much for your comment.
Round 3
Reviewer 1 Report
The authors have addressed all my questions.